# Seasonal Dynamics of Culturable Yeasts in Ornithogenically Influenced Soils in a Temperate Forest and Evaluation of Extracellular Enzyme Secretion in *Tausonia pullulans* at Different Temperatures

**DOI:** 10.3390/jof10080532

**Published:** 2024-07-30

**Authors:** Anna Glushakova, Anna Sharova, Aleksey Kachalkin

**Affiliations:** 1Soil Science Faculty, M.V. Lomonosov Moscow State University, 119991 Moscow, Russia; kachalkin_a@mail.ru; 2I.I. Mechnikov Research Institute of Vaccines and Sera, 105064 Moscow, Russia; 3G.K. Skryabin Institute of Biochemistry and Physiology of Microorganisms of RAS, 142290 Pushchino, Russia; 4Institute for African Studies of RAS, 123001 Moscow, Russia

**Keywords:** wild birds, psychrotolerant yeasts, pedobionts, esterases, lipases, proteases

## Abstract

The culturable yeast communities in temperate forest soils under the ornithogenic influence were studied in a seasonal dynamic. To investigate the intense ornithogenic influence, conventional and “live” feeders were used, which were attached to trees in the forest and constantly replenished throughout the year. It was found that the yeast abundance in the soil under strong ornithogenic influence reached the highest values in winter compared to the other seasons and amounted to 4.8 lg (cfu/g). This was almost an order of magnitude higher than the minimum value of yeast abundance in ornithogenic soils determined for summer. A total of 44 yeast species, 21 ascomycetes and 23 basidiomycetes, were detected in ornithogenic soil samples during the year. These included soil-related species (*Barnettozyma californica*, *Cyberlindnera misumaiensis*, *Cutaneotrichosporon moniliiforme*, *Goffeauzyma gastrica*, *Holtermanniella festucosa*, *Leucosporidium creatinivorum*, *L. yakuticum*, *Naganishia adeliensis*, *N. albidosimilis*, *N. globosa*, *Tausonia pullulans*, and *Vanrija albida*), eurybionts (yeast-like fungus *Aureobasidium pullulans*, *Debaryomyces hansenii*, and *Rhodotorula mucilaginosa*), inhabitants of plant substrates and litter (*Cystofilobasidium capitatum*, *Cys. infirmominiatum*, *Cys. macerans*, *Filobasidium magnum*, *Hanseniaspora uvarum*, *Metschnikowia pulcherrima*, and *Rh. babjevae*) as well as a group of pathogenic and opportunistic yeast species (*Arxiozyma bovina*, *Candida albicans*, *C. parapsilosis*, *C. tropicalis*, *Clavispora lusitaniae*, and *Nakaseomyces glabratus*). Under an ornithogenic influence, the diversity of soil yeasts was higher compared to the control, confirming the uneven distribution of yeasts in temperate forest soils and their dependence on natural hosts and vectors. Interestingly, the absolute dominant species in ornithogenic soils in winter (when the topsoil temperature was below zero) was the basidiomycetous psychrotolerant yeast *T. pullulans*. It is regularly observed in various soils in different geographical regions. Screening of the hydrolytic activity of 50 strains of this species at different temperatures (2, 4, 10, 15 and 20 °C) showed that the activity of esterases, lipases and proteases was significantly higher at the cultivation temperature. Ornithogenic soils could be a source for the relatively easy isolation of a large number of strains of the psychrotolerant yeast *T. pullulans* to test, study and optimize their potential for the production of cold-adapted enzymes for industry.

## 1. Introduction

In forest ecosystems, there are a variety of ecological niches that differ in the habitat conditions in which yeasts develop, and temperate forests (broadly defined as forests in the temperate zone that lie between boreal and tropical forests) are no exception. Yeast species from these ecological niches sooner or later “come into contact” with the organogenic soil horizon and participate in the formation of soil yeast complexes. Although yeasts in temperate forest soils have long been studied in detail, and much is known about their diversity and functional role in soil-related processes (nutrient transformation and maintenance of soil structure) [1,2,3,4,5,6], much remains unknown. The distribution of forest topsoil yeasts is uneven and is influenced by various parameters. Yurkov et al. [7] showed that in three temperate forests in Germany (in three regions), only one common basidiomycete species *Apiotrichum dulcitum*, was present. In three Mediterranean xerophytic forests sampled at a single locality, 8 of 57 species were found at all three sites [8]. The differences in species composition between sites lead to high diversity values at the regional level [8,9]. 

The structure of the yeast complexes depends on many biotic factors: the presence/absence of anthills [10], various invertebrate species and nests of insect larvae [11], the feces of migratory and resident birds [12,13,14], and the excrement of wild animals [15].

Recent studies have shown that quite well-analyzed soils yield a large number of undescribed yeasts. The proportion of potentially new taxa has been estimated to exceed 30% in temperate beech forests and Mediterranean xerophytic forests [8,16]. The same applies to some other temperate forests [1,17,18,19]. Therefore, further studies of soil yeasts in forest ecosystems are promising to find new species and expand the understanding of soil yeast diversity. In addition, a number of biotic and abiotic factors that can significantly influence yeasts in forest soils are still insufficiently analyzed. 

The discovery of soil yeast species that can not only grow but also increase their abundance during the cold period at subfreezing temperatures and the assessment of their ability to actively secrete hydrolases (proteases, lipases, esterases, amylases, cellulases and pectinases) could prove to be an important source for the relatively simple and stable isolation of a large number of strains that could subsequently be studied, tested and optimized for biotechnological purposes. Such isolates can potentially contribute to industrial processes that require high enzymatic activity at low temperatures, including the bread, baking, textile, food, biofuel, detergent and brewing industries. Perhaps the greatest advantage of using psychrophilic enzymes is the reduction in energy consumption and processing costs associated with the industrial heating steps [20]. 

In our study, we investigated the ability of one of the most abundant “winter” soil yeast species to secrete three cold-active hydrolases (esterase, lipase and protease) widely distributed among yeasts, which enable it to utilize complex substances as an energy source. In addition, esterase and lipase provide access to phospholipids, glycerol and fatty acids necessary for membrane maintenance under cold conditions, which is important for survival.

Little is also known about the influence of wild birds, an integral part of the forest ecosystem, on the yeast complexes in the topsoil through their droppings and their ability to increase the diversity of yeasts by a fraction. At the same time, it has been shown that the feces of migratory birds, wild birds and semi-synanthropic birds can contain specific and diverse yeast complexes and are also a source of new yeast species [13,14,21,22]. To our knowledge, ornithogenic soils have been studied around the nests of bird species in Antarctica [23,24,25,26]. No such studies are known for temperate forests.

The Meshcherskaya Lowland, where we conducted our study, is an extensive forest area in the center of the Eastern European Plain. The typical habitats of the temperate climate belt, the eastern part of the Atlantic continental forest climate zone, have remained practically untouched in this lowland area. More than 200 bird species live here, which can transmit various yeasts with their droppings, including opportunistic pathogenic species. 

The aim of this work was to study the culturable yeast diversity in a seasonal dynamic in ornithogenically influenced topsoils in a pristine mixed forest area, to observe and identify the most abundant cold-adapted yeast species and to study its esterase, lipase and protease extracellular enzyme activities at different temperatures.

## 2. Materials and Methods

### 2.1. Study Location, Sample Processing, Yeast Cultivation and Isolation

The study was conducted in the Meshcherskaya lowland in the southwestern sector of the Vladimir region (55°10′58″ N 40°20′00″ E). This is a zone of mixed forests in the center of the East European Plain in Russia (predominant species of the first stage are *Betula verrucosa*, *Picea abies*, *Pinus sylvestris*, *Populus tremula*; predominant species in the understory are *Corylus avellana*, *Euonymus verrucosus*, *Frangula alnus*, *Malus silvestris* and *Sorbus aucuparia*). The lowland has a temperate continental climate with frosty, relatively cold winters and warm, rarely hot summers (average temperature in January is 10 °C below zero, in July 18 °C above zero; annual precipitation 500 mm) (Figure 1, Table 1).

The soils are mainly podzolic, according to the World Reference Base for Soil Resources [27] (typical soils of forests formed in cold areas with good leaching). For the study of ornithogenically influenced soils in the forest, dynamic sampling was carried out throughout the year from April 2023 to April 2024. Twice a month, soil samples were taken from the topsoil (0–15 cm) under trees (*Betula verrucosa*) where forest bird feeders were installed. The samples were taken in the morning between 08:00 and 09:00 GTM to reduce the temperature fluctuations between the individual samples (they did not exceed 1 °C). For each sample, soil temperature was measured using a soil thermometer TP-2 (Klin, Russia); the hydrogen index (pH_H20_) was measured in situ with a HI981030 GroLine Soil pH Tester (Hanna Instruments, Scientific Park in Salaj county, Romania). The hydrogen index (pH_H20_) was between 3.6 and 5.4 for the control soil samples and between 3.1 and 4.2 for the ornithogenic soil samples. A total of five sites were investigated at a distance of 500–600 m from each other. At the sites, three wooden feeders (30 × 25 × 30 cm) and three “live” feeders were placed at a height of 2–2.5 m on a tree (Figure 2).

The composition of the bird feed consisted of wheat, oats, whole maize, crushed maize, barley, peanuts, sunflower seeds, linseed and red millet. Control soil samples were taken under trees without feeders at a 100–150 m distance from each tree with feeders. A total of 60 samples were taken per month (30 samples of ornithogenic soil and 30 samples of control soil). A total of 720 soil samples were analyzed over twelve months. In winter, when it snowed, and the drifts in the forest reached 40–60 cm, the drifts under the study trees were regularly plowed with sterile shovels. Care was taken to ensure that the snow cover did not exceed 2 cm so that bird droppings could be deposited evenly and abundantly in the soil. During the entire study period, 16 bird species (ornithologists employees of the Meshchera National Park helped us to observe and identify the bird species using Carl Zeiss Victory 8 × 42 SF (42 × 8) binoculars, Wetzlar, Germany) visited the feeding sites most frequently (Figure 3).

Samples were collected using sterile gloves and trowels. The collected samples were packed in a sterile zip bag and provided with an accompanying label. The samples were packed in special cooling bags at a temperature of 4 °C and delivered to the laboratory for microbiological analysis within 12 h. In the laboratory, the samples were stored in a cold chamber at a temperature of no more than 4 °C for a maximum of three days. Each soil sample was then taken and poured with sterile physiological saline solution to obtain a dilution of 1:10. The suspensions were vortexed on a Multi Reax Vortexer (Heidolph Instruments, Schwabach, Germany) for 15 min at 2000 rpm. Three suspensions were prepared for each sample. The prepared suspensions were plated (100 μL per plate) in three replicates each on GPY agar media (20 g/L glucose, 10 g/L peptone, 5 g/L yeast extract, 20 g/L agar) supplemented with chloramphenicol (500 mg/L). The plates were incubated at 20 °C for 9–12 days and checked regularly. As soon as a colony became visible, it was transferred to a fresh GPY agar plate. The colonies were differentiated into macromorphological types using a dissecting microscope, counted and 3–5 representatives of each colony type were transferred to a pure culture and then molecularly identified.

### 2.2. Molecular Identification (DNA Extraction, Amplification, Sequencing and Analysis)

The yeasts were molecularly identified using the ITS rDNA region as a universal DNA-barcoding for fungi [28]. The nuclear ribosomal ITS1-5.8S-ITS2 region was amplified and sequenced using ITS5 primer. The criteria described in Vu [29] were used to separate the yeast species. DNA isolation and PCR were performed according to the procedure described previously [30,31]. DNA sequencing was performed using the Big Dye Terminator V3.1 Cycle Sequencing Kit (Applied Biosystems, Waltham, MA, USA) with subsequent analysis of the reaction products on an Applied Biosystems 3130xl Genetic Analyzer at the facilities of Evrogen (Moscow, Russia). For sequencing, the ITS5 primer (5′-GGA AGT AAA AGT CGT AAC AAG G) was used [31]. For species identification, nucleotide sequences were compared with those in public databases, using the BLAST NCBI (www.ncbi.nlm.nih.gov (accessed on 24 May 2024)) and the MycoID (www.mycobank.org (accessed on 24 May 2024)) tools. The ITS regions of the strains studied were 99.5–100% similar to the type strains. Sequences obtained in the present study for yeast species were deposited in the GenBank database (PP905601–PP905645, PP481708, Table 2). All the purified and sequenced yeast strains isolated in this study were cryopreserved in 10% (*v*/*v*) glycerol in water solution at −80 °C in the yeast collection of the Soil Biology Department at Lomonosov Moscow State University (WDCM CCINFO number: 1173; catalog: https://depo.msu.ru/, accessed on 10 June 2024).

### 2.3. Evaluation of Esterase, Lipase and Protease Extracellular Enzyme Activities for Strains of Tausonia pullulans at 2, 4, 10, 15 and 20 °C

All isolates of *T. pullulans* examined for hydrolytic activity were first tested for their ability to grow at different temperatures (2, 4, 10, 15, 20, 25 and 28 °C) on GPY agar plates. Growth was visually monitored daily for 14 days. 100% of the isolated yeasts were able to grow in a temperature range between 2–25 °C and at 28 °C, no colony growth was observed. Thus, all investigated strains could be classified as psychrotolerant microorganisms [32,33].

For enzyme screening, yeast isolates were cultured in GPY medium with the addition of inducing substrates. Calibrated (by spectrophotometry) suspensions of 10^7^ cells/mL grown for 48 h were inoculated onto the surface of agar plates (10 µL).

For the analysis of esterase, plates with Tween-80 agar were used (10 g/L, 5 g/L NaCl, 0.10 g/L CaCl_2_·2H_2_O, pH 6.8); an opaque halo around a colony indicated esterase production, which was visualized by CaCl_2_ precipitation [34,35]. For the analysis of lipase, olive oil (4%) was added to the growth medium (pH 6.8) as an inducer together with Rhodamine B dye (0.01%), and UV light (350 nm) indicated the yellowish fluorescent halos [35]. Protease activity was assessed on GPA plates containing skimmed milk powder (2%, pH 6.6); a clear zone around a colony on the plate was indicative of protease activity [36]. The secretion ability was measured using a digital paquimeter and assessed (mm) as follows: ++, strongly positive, for values > 4.0; +, positive, for values between 2.0 and 4.0; w, weakly positive, for values between 1.0 and 2.0; w–, weakly negative, for values between 0.1 and 1.0, and –, negative, no clear zone [37]. The results were determined based on the average from three individual experiments for each strain. 

### 2.4. Statistical Data Analyses

The number of yeast colonies was used to calculate the abundance of yeast cells (cfu) in each type of sample per dry weight. The structure of the yeast community was determined for each sample. Relative abundance was calculated as the proportion (%) of a particular species in the sample and was based on the number of colonies. The species diversity of yeasts was estimated using the Shannon index [38]. Simpson’s diversity index (1 – D) was used to assess the dominance of yeast species [39]. Species evenness was assessed using the Pielou index [40]. The similarities among yeast groups from different soils were estimated using UMPGA clustering technique based on the Sorensen and Bray–Curtis indices. Similarity percentage (SIMPER) analysis was used to determine which species was responsible for driving the differences in community composition among groups. All clusterings and SIMPER analyses were performed using PAST 4.04 [41]. Statistical analyses were performed using Statistica 8 (StatSoft Inc., Tulsa, OK, USA) at two hierarchical levels of factors: condition of the soil (ornitogenically influenced soil and control samples) and season (summer, fall, winter, spring). The normality of the distribution of yeast numbers was tested for the variables discussed. Effects were considered statistically significant at the *p* ≤ 0.05 level. After the application of the Shapiro–Wilk test, analysis of variance (ANOVA) was performed to determine significant differences in the observation of hydrolytic enzyme activity in strains of *T. pullulans* at different temperatures and total yeast abundance.

## 3. Results

### 3.1. Yeast Abundance

The abundance of soil yeasts changed synchronously throughout the year, both in the ornithogenic soil and in the control. It reached a minimum in summer and a maximum in winter (Figure 4). However, the increase in abundance was more pronounced in the ornithogenic soil. Two-way ANOVA showed that the abundance of soil yeasts depended on the season of sampling (F = 149.02, α = 0.05) and on the condition of the soil (ornithogenically influenced/control) (F = 150.20, α = 0.05).

### 3.2. Yeast Diversity

A total of 46 yeast species were found in this study (both in ornithogenically influenced and control soils), including one potentially new species. They belong to four lineages of *Fungi*, *Pezizomycotina* (2 species), *Saccharomycotina* (20 species), *Agaricomycotina* (18 species) and *Pucciniomycotina* (6 species). A total of 44 yeast species (21 ascomycetes and 23 basidiomycetes) were detected in the ornithogenic soil; 38 yeast species (15 ascomycetes and 23 basidiomycetes) were observed in the control soil (Table 2). A total of six opportunistic and potentially pathogenic ascomycetous species were observed during the year: *A. bovina*, *C. albicans*, *C. parapsilosis*, *C. tropicalis*, *Cl. lusitaniae* and *N. glabratus*. Two of them (*C. parapsilosis* and *C. tropicalis*) were detected in both ornithogenic and control soil samples. However, the proportion of both species was significantly dependent on soil condition (F = 102.57 and 15.78, α = 0.05, for *C. parapsilosis* and *C. tropicalis*, respectively) and was higher in ornithogenic soil. The observed species richness in the ornithogenically influenced soil varied between the minimum number of 23 species in summer and the maximum number of 39 species in spring (in the control, the minimum number was observed in fall, 24 species, the maximum—also in spring, 33 species). In the ornithogenic soil, diversity and evenness ranged from the minimum in winter (H’ = 2.63, J’ = 0.49) to the maximum in spring (H’ = 3.05, J’ = 0.56) (in the control also from the minimum in winter (H’ = 2.79, J’ = 0.51) to the maximum in spring (H’ = 3.04, J’ = 0.56)) (Table 2).

### 3.3. Comparison of Yeast Groups

The comparison of the studied samples of ornithogenically influenced soils in a temperate forest and control samples in a seasonal dynamic using beta diversity similarity/dissimilarity measures of the relative abundance (by the Bray–Curtis index) and list (by Sorensen index) of yeast species showed that the yeast communities in ornithogenic soils differ from the control soils without the strong influence of birds (Figure 5). Differences in the structure of yeast complexes and yeast species numbers by season were observed. The significant effect of the ornithogenic load on the soils is evident when the structure of the yeast groups is compared. Using the Bray–Curtis index for clustering separated control and ornithogenic soils. The greatest similarity (Sorensen index) was found between summer and fall samples in both ornithogenic and control soils; winter and spring complexes were dissimilar and differed strongly from summer and fall (Figure 5).

SIMPER analysis revealed that three yeast species were responsible for approximately 50% of the differences in community composition between control and ornithogenic soils during all seasons. These species were *Tausonia pullulans*, *Candida zeylanoides* and *Cystofilobasidium capitatum*. *T. pullulans* made the greatest contribution to the differentiation of yeast groups between control and ornithogenic soils in all seasons except summer when *C. zeylanoides* and *Metschnikowia pulcherrima* were the main differentiators.

### 3.4. Production of Hydrolytic Enzymes

A total of 50 yeast isolates of *T. pullulans* obtained from ornithogenically influenced soils were analyzed for their esterase, lipase and protease activity at different temperatures (2, 4, 10, 15, 20 °C). For all three enzymes, the maximum enzymatic activity was observed at the minimum test temperature (plus 2 °C) and the minimum enzymatic activity at the maximum test temperature (plus 20 °C). Esterase secretion was most pronounced in the tested strains of *T. pullulans* at all tested temperatures, followed by lipase and protease. In our study, the secretion of hydrolytic enzymes was significantly dependent on temperature (F = 3445.63, 575.20, 362.62, α = 0.05 for esterases, proteases and lipases, respectively) (Figure 6, Appendix A).

## 4. Discussion

### 4.1. Yeast Abundance

An increase in the epiphytic yeast population in a temperate forest in winter has already been demonstrated for litter (*Betula verrucosa* Roth, *Quercus robur* L., *Tilia cordata* Mill.) and leaves of plants that overwinter with green leaves under snow (*Ajuga reptans* L., *Oxalis acetosella* L.) [42]. The same trend has been observed in epiphytic yeasts on different plant species [43]. In this study, the increase in the abundance of soil yeasts in winter was associated with the development of psychrophilic species, representatives of the genus *Leucosporidium*, and a significant increase in the proportion of *T. pullulans*. Most likely, such an increase in the abundance of the yeast *T. pullulans* is related to its adaptations to the selective pressure of the cold environment, with a high content of unsaturated fatty acids in the cell membrane, as well as the synthesis of enzymes active at low temperatures, which allows it to efficiently utilize the nutritional sources [44,45,46]. The boost in abundance was significantly higher in ornithogenically influenced soils, which is probably due to the fact that they contain high levels of organic carbon and nitrogen in the form of amino acids and urea [47].

### 4.2. Yeast Diversity

Studies on ornithogenic soils in Antarctica (colonies of marine bird species in coastal regions and on islands with exposed ice-free soil) have been carried out using both culturing methods [25] and metabarcoding [26]. *Tausonia pullulans*, *Candida*, *Glaciozyma antarctica*, *Holtermanniella wattica*, *Malassezia*, *Filobasidiella* and *Leucosporidium* were the taxa assigned by metabarcoding, with *T. pullulans* being among the most abundant [26]. de Sousa et al. [25] obtained strains of *Debaryomyces* sp., *Papiliotrema laurentii*, *Rhodotorula mucilaginosa* from ornithogenically influenced soils in Antarctica using culturing methods with *Rh. mucilaginosa* being among the most abundant species; *Candida glaebosa* and *Debaryomyces macquariensis* were also observed in ornithogenic soils (penguin soils) using culturing methods [23,24]. In our study of ornithogenic soils in a temperate forest in seasonal dynamics using the culturing method, 44 yeast species were detected. Species richness was highest in the ornithogenic soil and in the control in spring, with 39 and 33 species, respectively. In the ornithogenic soil, this was due to the detection of pathogenic and opportunistic ascomycetous yeasts *A. bovina*, *C. albicans*, *Cl. lusitaniae*, *N. glabratus*, which were never found in the control, and soil-related species of the genus *Naganishia* (*N. albida*, *N. albidosimilis* and *N. globosa*), which were detected both in the ornithogenic soil and in the control. It is possible that yeast species whose physiological activity was suppressed in winter became more active again in spring (Table 2). Two opportunistic *Candida* species were found not only in the ornithogenically influenced soil but also in the control during the entire study year. These were the yeasts *C. parapsilosis* and *C. tropicalis*. Thus, among other things, birds transmit different clinically important yeasts and can contaminate soil and water sources.

In winter, when the soil temperature was below zero, the proportion of basidiomycetous psychrophilic yeasts of the genus *Leucosporidium* (*L. creatinivorum*, *L. intermedium* and *L. yakuticum*) and the species *Sampaiozyma ingeniosa* increased in the ornithogenic soil (and in the control). The relative abundance of species of the genus *Leucosporidium* was significantly dependent on the season (F = 15.13, 13.56 and 35.58 α = 0.05, for *L. creatinivorum*, *L. intermedium* and *L. yakuticum*, respectively). The yeasts *L. intermedium*, *L. yakuticum* and *L. creatinivorum* were also regularly found in Antarctic and Arctic soils [48,49,50,51,52]. It is particularly noteworthy that the proportion of the psychrophilic yeast *T. pullulans* increased significantly during the winter. While it reached 15% in the control, it was almost 30% in the ornithogenic soil (while in summer, it was 2% and 10%, respectively). This is the only yeast species whose relative abundance was significantly dependent on both season (F = 32.12, α = 0.05) and soil condition (ornithogenic/control) (F = 28.06, α = 0.05).

*T. pullulans* is a basidiomycetous yeast from the order *Cystofilobasidiales* (*Agaricomycotina* and *Tremellomycetes*) [53]. It is a widespread, psychotolerant, soil-related species that was first described from the atmosphere in 1901 as *Oidium pullulans* Lindner (type strain CBS 2532). *T. pullulans* strains were found in soils on Scott Base (Ross Island) [54], on East Ongul Island, East Antarctica [37]; in the province of Tierra del Fuego (Argentina, Antarctica) [55]; in Patagonian forest soils [18]; in soils from the sub-Antarctic region [20]; in European glaciers [56]; in Arctic habitats (plants and soils) [26,47,57]; in soils near fruit trees in Slovakia [58]; in ornithogenically influenced maritime Antarctic soils [26]. In addition to soil, it has also been isolated from spring fluxes (xylem sap leaking from cuts on limbs and trunks resulting from winter damage caused by freeze–thaw cycles and injuries caused by birds and animals) of trees in Braunschweig, Lower Saxony, Germany [59]; it has been found on food (in open packaging in kimchi at 4 °C) [60]. In our study, T. pullulans was the most abundant taxa in winter, emphasizing its ability to successfully adapt and survive in the hostile conditions of sub-freezing soil temperatures. The ornithogenically influenced soils in a temperate forest in winter (when the topsoil temperature is below zero) could probably be a proven site for the active reproduction of the basidiomycetous psychrotolerant yeast *T. pullulans*. On average, this species is considered a permanent but minor component of the pedobiont yeast community in soils of the intracontinental temperate climate zone [1].

Data derived from the diversity characterization in several studies indicate that basidiomycetous yeasts are better adapted to cold environments than ascomycetous yeasts [61]. However, in some studies, yeasts from the phylum *Ascomycota* have been isolated more frequently [26]. Recently, it has been shown that the genomes of psychrophilic yeasts of the phylum *Basidiomycota* contain more gene clusters for the synthesis of secondary metabolites than those of the phylum *Ascomycota*. At the same time, the genome size of the psychrophilic yeasts of the phylum *Basidiomycota* is larger than that of the phylum *Ascomycota*. The psychrophilic yeasts of the phylum *Basidiomycota* also encode more catalytic enzymes and may therefore be more environmentally tolerant [62].

Other yeast species found during the year in both ornithogenic soil and the control included typical ascomycetous (*B. californica*, *C. sake* and *Cyb. misumaiensis*) and basidiomycetous (*Cut. moniliiforme*, *G. gastrica* and *V. albida*) soil-related yeasts [1,4,63,64,65,66,67,68,69]; characteristic species not only of soil but also of forest litter, tree sap and aquatic habitats (*Cys. capitatum*, *Cys. Infirmominiatum* and *Cys. macerans*) [1,70,71,72,73,74,75]; epiphytes (*F. magnum*, *H. uvarum*, *P. flavescens* and *Rh. babjevae*) [43,68,73,76,77]; eurybiont species (yeast-like fungus *Aur. pullulans*, *D. hansenii* and *Rh. mucilaginosa*) [67]. We would also like to draw attention to the ascomycetous yeast *C. zeylanoides* found. The relative abundance of this species was significantly dependent on the soil condition (F = 219.92, α = 0.05) and was higher in ornithogenic soils (for *C. zeylanoides*, about 10% regardless of the season, in the control, it did not exceed 1.5%). Previously, the yeast *C. zeylanoides* was found in fresh feces of the partially synanthropic birds *Bombycilla garrulus* (Bohemian waxwing) and *Pyrrhula pyrrhula* (Eurasian bullfinch) [22]. It can be cautiously assumed that this species could be an intestinal symbiont of some other wild bird species.

### 4.3. Extracellular Enzyme Secretion by T. pullulans

The ability of yeasts to grow and multiply in cold environments suggests that their metabolism is catalyzed by enzymes that are active at low temperatures. *T. pullulans* is a psychrotolerant yeast with several extracellular enzymatic activities. It can play an important role in the decomposition of organic matter, nutrient cycling and fertilization of soil in the winter season. Its biotechnological potential includes the production of cold-adapted proteases, lipases, esterases, amylases, cellulases and pectinases [78]. It has even shown pronounced lipase secretion at subfreezing temperatures [37]. In our study, we decided to extend the knowledge of the ability to secrete three hydrolases (esterase, lipase and protease) in this species.

In this context, strains of the most abundant yeast species in the winter season, *T. pullulans*, isolated from ornithogenically influenced soils, were subjected to screening of different enzymes at low and moderate temperatures.

The lower the cultivation temperature, the higher the activity. The highest activity was observed at plus 2 °C, the minimum at plus 20 °C. The secretion of proteases was no longer observed at plus 20 °C. We also examined the secretion of these three enzymes at plus 25 °C, but we obtained negative results for all three hydrolytic enzymes.

Although this yeast is exposed to temperatures below zero from late fall to early spring, which is detrimental to its survival, it can still grow and increase its abundance at low temperatures. The secretion of extracellular enzymes enables it to utilize complex substances as an energy source [79,80]. Esterases and lipases provide access to phospholipids, glycerols and fatty acids, which are necessary for maintaining the fluidity of the cell membrane in the cold conditions of the winter season, which is essential for their survival [81,82]. Strains of *T. pullulans* have already been reported to produce a variety of cold-active exoenzymes [24,37,55,80]. It has also been shown that maximum lipase activity is observed at minus 3 °C [37].

Enzymes active at low temperatures from strains of the culturable yeast species *T. pullulans*, which can be detected relatively easily and in high abundance in ornithogenic soils, could be used in various biotechnological processes in different industries as well as in environmental applications in processes for the bioremediation of pollutants [83]. In addition, esterases could be used as diagnostic reagents for measuring cholesterol in human blood serum [84,85].

## 5. Conclusions

The study of the seasonal dynamics of the abundance of culturable yeasts in the topsoil under the ornithogenic influence in temperate forests showed that the maximum values occurred in winter when the average soil temperature was below zero. Abundance increased mainly due to the high proportion of the psychrotolerant yeast *T. pullulans*. The study of the ability of strains of this species to produce hydrolytic enzymes (esterase, lipase and protease) at different temperatures (2, 4, 10, 15 and 20 °C) showed that the maximum activity occurred at the lowest temperature of the study (2 °C). It was found that the diversity of yeasts was higher in ornithogenic soils than in the control. This was mainly due to the detection of a group of pathogenic and opportunistic species. Six such species were found in the ornithogenic soils: *A. bovina*, *C. albicans*, *C. parapsilosis*, *C. tropicalis*, *Cl. lusitaniae* and *N. glabratus*, whereas only two species, *C. parapsilosis* and *C. tropicalis*, were found in the control soil.

## Figures and Tables

**Figure 1 jof-10-00532-f001:**
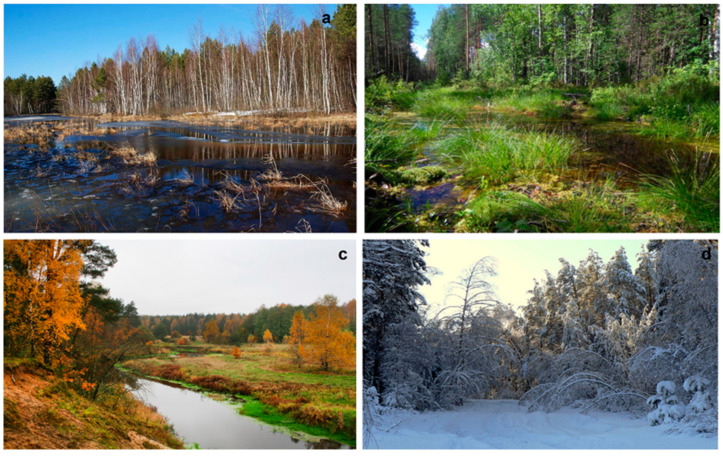
Four seasons in Meshcherskaya lowland (southwestern sector of the Vladimir region, 55°10′58″ N 40°20′00″ E); (**a**) Spring; (**b**) Summer; (**c**) Fall; (**d**) Winter.

**Figure 2 jof-10-00532-f002:**
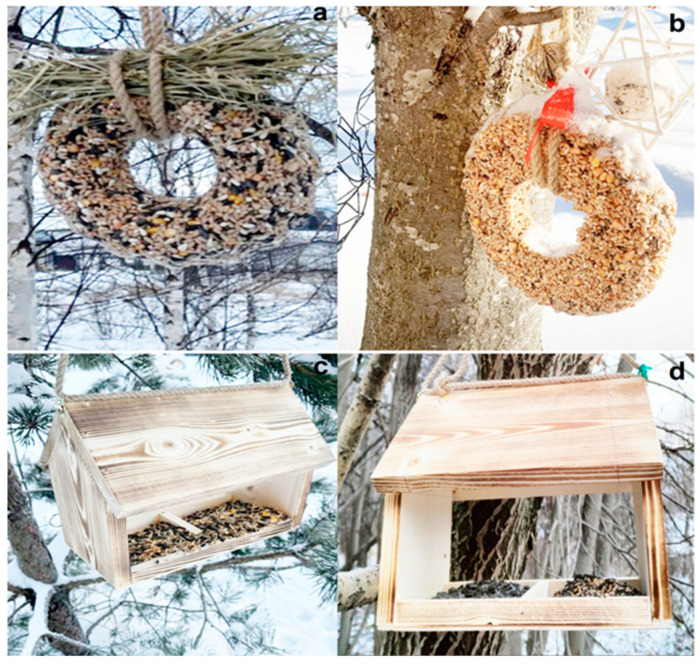
Examples of so-called “live” (**a**,**b**) and traditional (**c**,**d**) feeders.

**Figure 3 jof-10-00532-f003:**
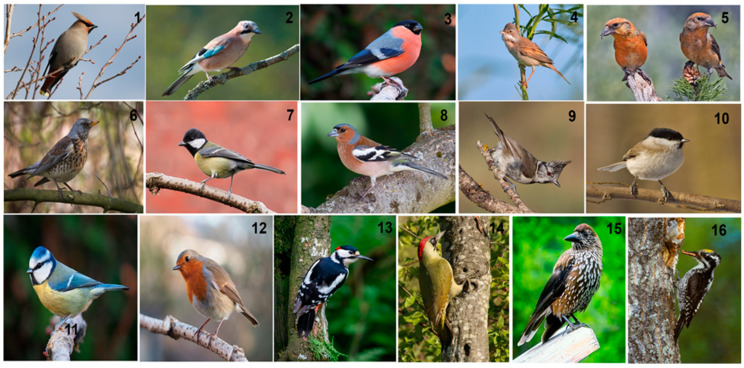
Forest bird species that visited the feeding sites most frequently in the year under observation: **1**—*Bombycilla garrulus* (the Bohemian waxwing); **2**—*Garrulus glandarius* (the Eurasian jay); **3**—*Pyrrhula pyrrhula* (the Eurasian bullfinch or bullfinch); **4**—*Curruca communis* (the common whitethroat or greater whitethroat); **5**—*Loxia curvirostra* (the red crossbill or common crossbill); **6**—*Turdus pilaris* (the fieldfare); **7**—*Parus major* (the great tit); **8**—*Fringilla coelebs* (the Eurasian chaffinch or chaffinch); **9**—*Parus cristatus* (the crested tit or European crested tit); **10**—*Poecile palustris* (the marsh tit); **11**—*Cyanistes caeruleus* (the Eurasian blue tit); **12**—*Erithacus rubecula* (the European robin); **13**—*Dendrocopos major* (the great spotted woodpecker); **14**—*Picus viridis* (the European green woodpecker); **15**—*Nucifraga caryocatactes* (the Eurasian nutcracker or nutcracker); **16***—Picoides tridactylus* (the Eurasian three-toed woodpecker). Photos were taken from the website https://en.wikipedia.org/ (accessed on 31 May 2024).

**Figure 4 jof-10-00532-f004:**
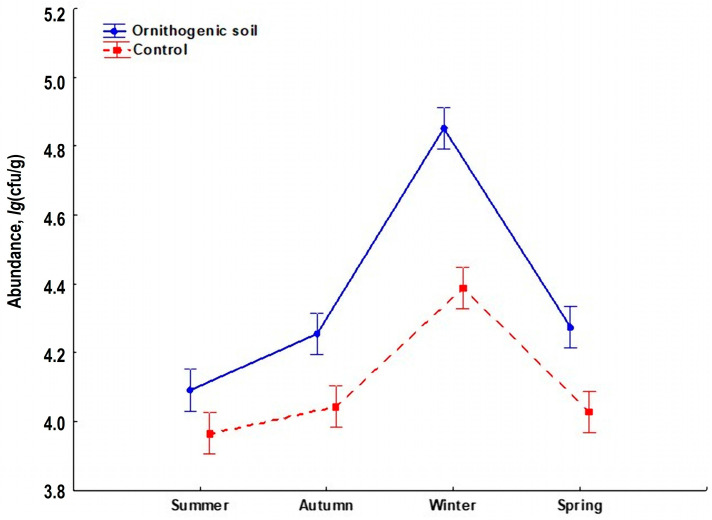
The seasonal dynamic of the abundance of culturable yeasts in ornithogenically influenced soil in a temperate forest (Meshcherskaya lowland, southwestern sector of the Vladimir region, 55°10′58″ N 40°20′00″ E).

**Figure 5 jof-10-00532-f005:**
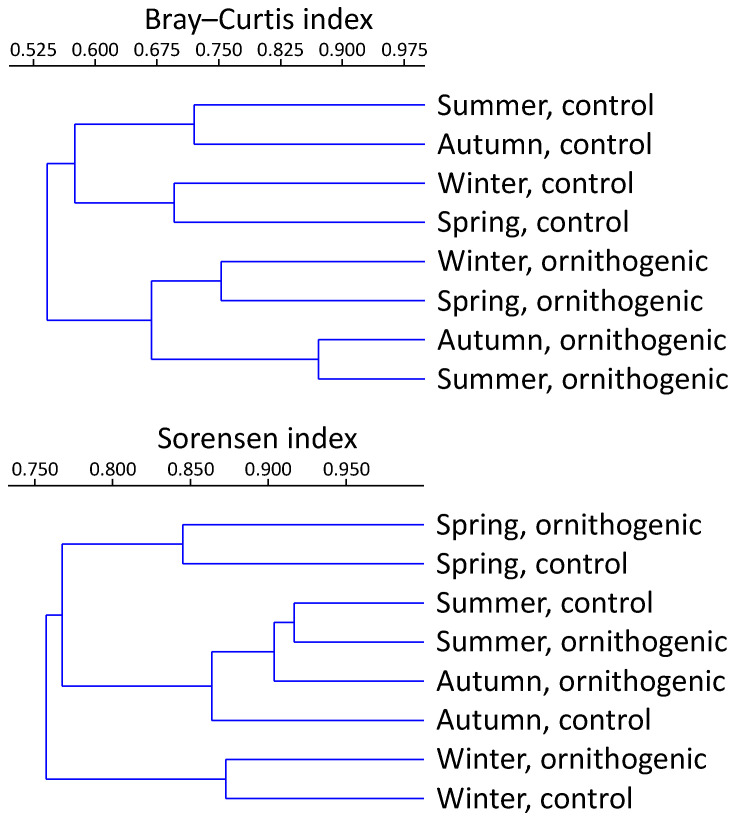
Comparison of studied yeast groups by clustering analysis using the Bray–Curtis and Sorensen measures based on relative abundances and lists of species. Meshcherskaya lowland, southwestern sector of the Vladimir region, 55°10′58″ N 40°20′00″ E, podzolic soil in a temperate forest).

**Figure 6 jof-10-00532-f006:**
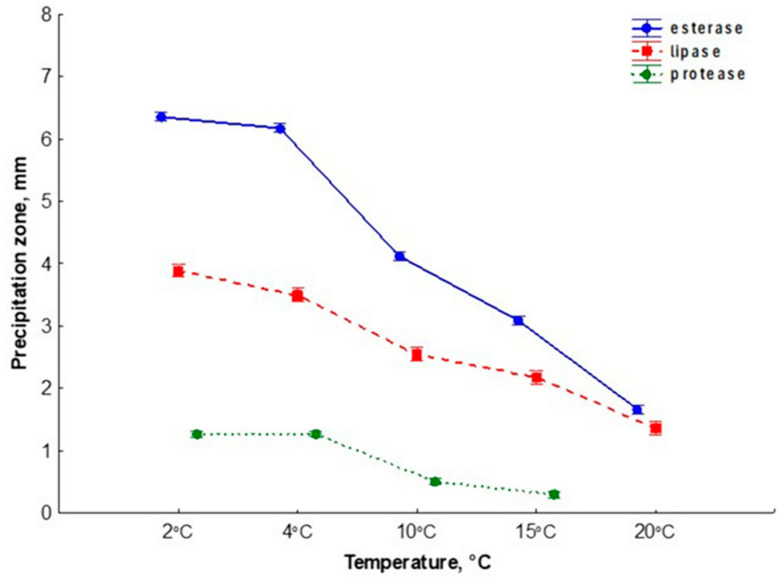
Secretion of esterase, lipase and protease in strains of the yeast *T. pullulans* at different temperatures (2, 4, 10, 15, 20 °C). Growth was visually monitored daily for 14 days.

**Table 1 jof-10-00532-t001:** Average monthly values of air and soil temperature at the location (Meshcherskaya lowland, southwestern sector of the Vladimir region, 55°10′58″ N 40°20′00″ E) during the sampling period.

Month	Air Day (°C)	Air Night (°C)	Topsoil (°C)
April	13	4	2
May	17	8	6
June	19	9	12
July	22	14	13.5
August	24	15	16
September	19	10	6
October	5	3	4
November	1	−2	−0.5
December	−5	−9	−4.5
January	−7	−8	−5.5
February	−4	−6	−5
March	3	−2	−3

**Table 2 jof-10-00532-t002:** Species list and relative abundance of yeast taxa isolated from the ornithogenically influenced soil in seasonal dynamic *. The indices of species richness, diversity and community evenness can be found below.

	* Relative abundance range in percentage value (%).					
Ornithogenic soil						
Control soil						
	0–5	5–10	10–15	15–20	20–25	25–30
**Yeast Species**		**GenBank Accession no.**	**Summer**	**Autumn**	**Winter**	**Spring**
	*Ascomycota*					
*Arxiozyma bovina* (Kurtzman & Robnett) Q.M. Wang, Yurkov & Boekhout		PP905601				

*Aureobasidium pullulans* (de Bary) G. Arnaud		PP905602				

*Barnettozyma californica* (Lodder) Kurtzman, Robnett & Basehoar-Power		PP905603				

*Candida albicans* (C.P. Robin) Berkhout		PP905604				

*Candida parapsilosis* (Ashford) Langeron & Talice		PP905605				

*Candida sake* (Saito & Oda) van Uden & H.R. Buckley		PP905606				

*Candida santamariae* Montrocher		PP905607				

*Candida tropicalis* (Castell.) Berkhout		PP905608				

*Candida zeylanoides* (Castell.) Langeron & Guerra		PP905609				

*Clavispora lusitaniae* Rodr. Mir.		PP905610				

*Cyberlindnera misumaiensis* (Y. Sasaki & Tak. Yoshida ex Kurtzman) Minter		PP905611				

*Debaryomyces hansenii* (Zopf) Lodder & Kreger-van Rij		PP905612				

*Debaryomyces fabryi* M. Ota		PP905613				

*Dothiora* sp.		PP905614				

*Hanseniaspora uvarum* (Niehaus) Shehata, Mrak & Phaff ex M.T. Sm.		PP905615				

*Metschnikowia pulcherrima* Pitt & M.W. Mill.		PP905616				

*Meyerozyma guilliermondii* (Wick.) Kurtzman & M. Suzuki		PP905617				

*Nakaseomyces glabratus* (H.W. Anderson) Sugita & M. Takash.		PP905618				

*Starmerella vitis* Čadež, Lachance, Drumonde-Neves, Sipiczki & G. Péter		PP905619				

*Yamadazyma mexicana* (M. Miranda, Holzschu, Phaff & Starmer) Billon-Grand (1989)		PP905620				

*Yarrowia alimentaria* (Knutsen, V. Robert & M.T. Sm.) Gouliam., R.A. Dimitrov, M.T. Sm. & M. Groenew.		PP481708				

*Yarrowia lipolytica* (Wick., Kurtzman & Herman) Van der Walt & Arx		PP905621				

	*Basidiomycota*					
*Cutaneotrichosporon moniliiforme* (Weigmann & A. Wolff) Xin Zhan Liu, F.Y. Bai, M. Groenew. & Boekhout		PP905622				

*Cystofilobasidium capitatum* (Fell, I.L. Hunter & Tallman) Oberw. & Bandoni		PP905623				

*Cystofilobasidium infirmominiatum* (Fell, I.L. Hunter & Tallman) Hamam., Sugiy. & Komag.		PP905624				

*Cystofilobasidium macerans* J.P. Samp.		PP905625				

*Filobasidium magnum* (Lodder & Kreger-van Rij) Xin Zhan Liu, F.Y. Bai, M. Groenew. & Boekhout		PP905626				

*Goffeauzyma gastrica* (Reiersöl & Di Menna) Xin Zhan Liu, F.Y. Bai, M. Groenew. & Boekhout		PP905627				

*Holtermanniella festucosa* (Golubev & J.P. Samp.) Libkind, Wuczk., Turchetti & Boekhout		PP905628				

*Kwoniella pini* (Golubev & Pfeiffer) Xin Zhan Liu, F.Y. Bai, M. Groenew. & Boekhout		PP905629				

*Leucosporidium creatinivorum* (Golubev) M. Groenew. & Q.M. Wang		PP905630				

*Leucosporidium intermedium* (Nakase & M. Suzuki) M. Groenew. & Q.M. Wang		PP905631				

*Leucosporidium yakuticum* (Golubev) M. Groenew. & Q.M. Wang		PP905632				

*Naganishia adeliensis* (Scorzetti, I. Petrescu, Yarrow & Fell) Xin Zhan Liu, F.Y. Bai, M. Groenew. & Boekhout		PP905633				

*Naganishia albida* (Saito) Xin Zhan Liu, F.Y. Bai, M. Groenew. & Boekhout		PP905634				

*Naganishia albidosimilis* (Vishniac & Kurtzman) Xin Zhan Liu, F.Y. Bai, M. Groenew. & Boekhout		PP905635				

*Naganishia diffluens* (Zach) Xin Zhan Liu, F.Y. Bai, M. Groenew. & Boekhout		PP905636				

*Naganishia globosa* Goto		PP905637				

*Naganishia vaughanmartiniae* Turchetti, Blanchette & Arenz ex Yurkov		PP905638				

*Papiliotrema flavescens* (Saito) Xin Zhan Liu, F.Y. Bai, M. Groenew. & Boekhout		PP905639				

*Rhodotorula babjevae* (Golubev) Q.M. Wang, F.Y. Bai, M. Groenew. & Boekhout		PP905640				

*Rhodotorula mucilaginosa* (A. Jörg.) F.C. Harrison		PP905641				

*Sampaiozyma ingeniosa* (Di Menna) Q.M. Wang, F.Y. Bai, M. Groenew. & Boekhout		PP905642				

*Tausonia pullulans* (Lindner) Xin Zhan Liu, F.Y. Bai, M. Groenew. & Boekhout		PP905643				

*Trichosporon aquatile* L.R. Hedrick & P.D. Dupont		PP905644				

*Vanrija albida* (C. Ramírez) M. Weiß		PP905645				

Species richness (ornithogenic soil/control)			23/25	28/24	30/25	39/33
Shannon index, H’ (ornithogenic soil/control)			2.74/2.87	2.79/2.82	2.63/2.79	3.05/3.04
Pielou index, J’ (ornithogenic soil/control)			0.51/0.52	0.51/0.52	0.49/0.51	0.56/0.56
Simpson (1-D) (ornithogenic soil/control)			0.91/0.93	0.91/0.93	0.88/0.92	0.93/0.94

## Data Availability

The datasets used and/or analyzed during the current study are available from the corresponding author upon reasonable request.

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
