# Peer review of "Seasonal Dynamics of Culturable Yeasts in Ornithogenically Influenced Soils in a Temperate Forest and Evaluation of Extracellular Enzyme Secretion in Tausonia pullulans at Different Temperatures"

_jof, 2024, doi:10.3390/jof10080532_

Round 1

Reviewer 1 Report

This manuscript describes the seasonal dynamics of culturable yeast in the topsoil under the ornithogenic influence in temperate forests. Several studies have been published on the Antarctic and Arctic. However, I found no similar previous research related to this manuscript. Therefore, this manuscript is of scientific importance in mycology. I also did not find any problems in this manuscript. Therefore, I recommend accepting this manuscript in JoF.

No.

Author Response

First of all, we would like to thank the Reviewer very much for the assessment, work and time.

Reviewer 2 Report

This study evaluated the diversity of culturable yeast in different soils and the qualitative evaluation of enzymes by T. pullulans in cold temperatures. Some results are interesting, but the manuscript exhibits flaws in the choice of statistical methods, the introduction could be improved, and the conclusion requires rewriting. Below, I addressed some additional comments.

Ln 47-54- It could be presented further down in the introduction. The introduction could start with a broader context rather than immediately discussing a specific yeast genus.

Ln 60-73- The paragraph does not add much to the introduction and would be more appropriate in the discussion section.

Ln 74- The introduction should start here!

Ln 113- It is mentioned that the extracellular enzymatic profile will be evaluated; however, only three enzymes were assessed. These enzymes were not mentioned in any part of the introduction, nor is it explained why these specific enzymes (esterase, protease, and lipase) were chosen for evaluation. Also, only one species was evaluated and it must be clear here.

Ln 230- The PCA is a parametric analysis and should not be applied to discrete data like abundance data. Instead, other analyses like PCoA or NMDS, with appropriate distance metrics (such as Bray-Curtis), should be used.

Ln 233- Once again, the statistical test used, ANOVA, is inappropriate for the results being evaluated, and non-parametric statistics should be adopted. Also, a proportion test, such as the chi-square test, could be used instead.

Ln 356- 44 yeast species or 46, as mentioned in Ln 250?

Ln 457- The conclusion looks more like an abstract! redirect the focus from a detailed summary and repetition of the results towards highlighting the most significant discovery or central outcome of the study. This is the section where authors should underscore the primary research finding or novel hypothesis that culminates and ties together the entire study, thereby concluding the narrative effectively. Think about it!

Figures and Tables

The table 2 is too large and could be compacted to provide better visualization.

Since the sequencing of the isolates was performed, it would be beneficial to include a phylogenetic analysis.

Minor comments:

Only the genus or species name should be italicized, while other taxonomic levels should not be in italics.

Author Response

First of all, we would like to thank the Reviewer very much for valuable comments, work and time.

We tried to do all our best to improve the manuscript.

Reviewer 2

This study evaluated the diversity of culturable yeast in different soils and the qualitative evaluation of enzymes by T. pullulans in cold temperatures. Some results are interesting, but the manuscript exhibits flaws in the choice of statistical methods, the introduction could be improved, and the conclusion requires rewriting. Below, I addressed some additional comments.

Ln 47-54- It could be presented further down in the introduction. The introduction could start with a broader context rather than immediately discussing a specific yeast genus. The Introduction has been corrected.

Ln 60-73- The paragraph does not add much to the introduction and would be more appropriate in the discussion section. The Introduction has been corrected.

Ln 74- The introduction should start here! We have started the Introduction here.

Ln 113- It is mentioned that the extracellular enzymatic profile will be evaluated; however, only three enzymes were assessed. These enzymes were not mentioned in any part of the introduction, nor is it explained why these specific enzymes (esterase, protease, and lipase) were chosen for evaluation. Also, only one species was evaluated and it must be clear here. The information has been added to the Introduction. We decided to investigate the most abundant detected cold-adapted yeast species (Tausonia pullulans) and to test more strains of one species for their extracellular enzymatic activity at different temperatures.

Ln 230- The PCA is a parametric analysis and should not be applied to discrete data like abundance data. Instead, other analyses like PCoA or NMDS, with appropriate distance metrics (such as Bray-Curtis), should be used. The approach to comparing yeast groups has been modified. Clustering based on the Sorensen and Bray-Curtis indices was used for comparison.

 Ln 233- Once again, the statistical test used, ANOVA, is inappropriate for the results being evaluated, and non-parametric statistics should be adopted. Also, a proportion test, such as the chi-square test, could be used instead. ANOVA is used to analyses total abundance of yeasts and enzymes activity.

Ln 356- 44 yeast species or 46, as mentioned in Ln 250? A total of 46 yeast species were found in this study (both in ornithogenically influenced and in control soils), and 44 species were detected in ornithogenic soil. The sentence has been corrected.

Ln 457- The conclusion looks more like an abstract! redirect the focus from a detailed summary and repetition of the results towards highlighting the most significant discovery or central outcome of the study. This is the section where authors should underscore the primary research finding or novel hypothesis that culminates and ties together the entire study, thereby concluding the narrative effectively. Think about it! Corrected.

Figures and Tables

The table 2 is too large and could be compacted to provide better visualization. We agree that the table is very large and unpleasant to read. However, we believe that it contains very important information about strains. All the strains are preserved in the collection and thus can be used for future studies. We have moved the table to the Supplementary Material.

Since the sequencing of the isolates was performed, it would be beneficial to include a phylogenetic analysis. Phylogeny is necessary to determine the taxonomic position of a species, but 99.5-100% similarity of the ITS rDNA region to the type strains allows for a clear species identity and taxonomic position.

Minor comments:

Only the genus or species name should be italicized, while other taxonomic levels should not be in italics. As recommended (https://doi.org/10.1186/s43008-020-00048-6), we have decided to italicize all Latin names. However, if this is not allowed by the rules of the journal, we will remove the additional italics.

With respect and gratitude,

authors

Reviewer 3 Report

General comments:

This paper investigates the seasonal dynamics of yeasts in temperate forest soils influenced by birds and the extracellular enzyme secretion of Tausonia pullulans at different temperatures. The study found that the abundance and diversity of yeasts in bird-affected soils significantly increase during winter, and that Tausonia pullulans exhibits high enzyme activity at low temperatures. These findings suggest that bird activity and seasonal changes have a significant impact on the composition and function of soil yeast communities. Additionally, the high enzyme production at low temperatures indicates that Tausonia pullulans has potential for industrial enzyme production under cold conditions.

Main comments:

1. The introduction section does not describe previous contributions or perspectives in this field, which detracts from highlighting the innovation of this paper, especially regarding the distribution of yeasts and their relationships with other organisms.

2. The study site is relatively small, and the paper does not explain the representativeness of this site for the research question.

3. While the results section analyzes the activity of yeasts and their enzymes, the discussion section does not elaborate on their ecological significance. For example, the paper finds that Tausonia pullulans has the highest enzyme activity at 2°C but does not discuss the underlying reasons and mechanisms.

Minor comments:

1. The description of Figure 5 is not detailed enough, which affects the reader's understanding of the information presented.

2. In line 94, the sentences "For example, seasonality and the associated change in topsoil temperature to below zero. Microbiological studies dealing with the identification and characterization of cold-adapted yeasts mainly focus on Arctic and Antarctic regions." are somewhat redundant.

3. In line 137, "30x25x30" should include units

Author Response

First of all, we would like to thank the Reviewer very much for valuable comments, work and time.

We tried to do all our best to improve the manuscript.

Reviewer 3

General comments:

This paper investigates the seasonal dynamics of yeasts in temperate forest soils influenced by birds and the extracellular enzyme secretion of Tausonia pullulans at different temperatures. The study found that the abundance and diversity of yeasts in bird-affected soils significantly increase during winter, and that Tausonia pullulans exhibits high enzyme activity at low temperatures. These findings suggest that bird activity and seasonal changes have a significant impact on the composition and function of soil yeast communities. Additionally, the high enzyme production at low temperatures indicates that Tausonia pullulans has potential for industrial enzyme production under cold conditions.

Main comments:

  1. The introduction section does not describe previous contributions or perspectives in this field, which detracts from highlighting the innovation of this paper, especially regarding the distribution of yeasts and their relationships with other organisms. Corrected.
  2. The study site is relatively small, and the paper does not explain the representativeness of this site for the research question. Information about the site has been added.
  3. While the results section analyzes the activity of yeasts and their enzymes, the discussion section does not elaborate on their ecological significance. For example, the paper finds that Tausonia pullulans has the highest enzyme activity at 2°C but does not discuss the underlying reasons and mechanisms.

Thank you for your comment. We have added the ecological significance of the psychrotolerant yeast species T. pullulans to the discussion: “It can play an important role in the decomposition of organic matter, nutrient cycling and fertilization of soil in winter season”. Unfortunately, we have not investigated the mechanisms and their genetic aspects. In our humble opinion, this requires a separate large study. However, the data recently obtained by researchers on genetic traits of the psychrophylic species could also explain to some extent the characteristics of the psychrotolerant species of T. pullulans: “ The genomes of psychrophilic yeasts of the class Basidiomycetes contained more secondary metabolite synthesis gene clusters than the class Ascomycetes. Meanwhile, the genome size of psychrophilic yeasts of the class Basidiomycetes was greater than that of the class Ascomycetes. The psychrophilic yeasts of class Basidiomycetes also encoded more catalytic enzymes and may therefore be more environmentally tolerant (Liu et al. 2023).”

Detail comments

Minor comments:

  1. The description of Figure 5 is not detailed enough, which affects the reader's understanding of the information presented. The figure and its description have been completely redesigned and corrected.
  2. In line 94, the sentences "For example, seasonality and the associated change in topsoil temperature to below zero. Microbiological studies dealing with the identification and characterization of cold-adapted yeasts mainly focus on Arctic and Antarctic regions." are somewhat redundant. The sentences have been deleted.
  3. In line 137, "30x25x30" should include units. Information has been added.

With respect and gratitude,

authors

Reviewer 4 Report

The topic addressed and the information presented in the manuscript is novel, highly relevant in ecological terms, in terms of knowledge of fungi and even in terms of the impact of birds as dispersers of fungi that could have an impact on public health. So the manuscript is according to the journal. I consider that the manuscript should be published after the authors consider and, where appropriate, address some aspects highlighted in this review.

The aspects observed that could contribute to improving the presentation of this interesting study are noted throughout the manuscript. Please review them in the attached file. In general terms I think that the introduction could be slightly complemented, the methods could be better detailed and some few aspects of the results-discussion improved. The conclusions should be presented in terms of highlighting the contributions and not as an extended discussion. Please see my comments throughout the manuscript.

My comments are indicated in detail in specific lines marked in yellow throughout the text. Please see the attached pdf file.

Author Response

First of all, we would like to thank the Reviewer very much for valuable comments, work and time.

Reviewer 4

The topic addressed and the information presented in the manuscript is novel, highly relevant in ecological terms, in terms of knowledge of fungi and even in terms of the impact of birds as dispersers of fungi that could have an impact on public health. So the manuscript is according to the journal. I consider that the manuscript should be published after the authors consider and, where appropriate, address some aspects highlighted in this review.

The aspects observed that could contribute to improving the presentation of this interesting study are noted throughout the manuscript. Please review them in the attached file. In general terms I think that the introduction could be slightly complemented, the methods could be better detailed and some few aspects of the results-discussion improved. The conclusions should be presented in terms of highlighting the contributions and not as an extended discussion. Please see my comments throughout the manuscript.

We have tried to do all our best to improve the article taking into account all the remarks and comments in the attached file.

With respect and gratitude,

authors

Round 2

Reviewer 2 Report

Unfortunately, I cannot recommend the work for publication

No comments

Author Response

Many thanks to the Reviewer for the work, the important comments and the time. Thanks to your work we were able to improve the manuscript.

Comment: The justification for selecting the enzymes evaluated continues not to be presented appropriately.

Response: We have tried to better justify the selection of these enzymes and to supplement the Introduction with information (Lines 77-85).

Comment:The statistical analyses applied are inappropriate.

Response:

The Fisher’s test was used for comparing average values after determining normality of distribution of data using the Shapiro–Wilk test.  Bray-Curtis index is based on abundance data and the Sorensen index is based on presence/absence data. These are the two approaches for estimating beta diversity in ecology, which together or separately have been used many times before, including yeast study. For examples, see: https://doi.org/10.1016/j.pedobi.2022.150822 https://doi.org/10.1007/s11557-024-01966-0 http://dx.doi.org/10.3389/fmicb.2021.637430 https://doi.org/10.1007/s12237-024-01377-0

Comment:The results are very exploratory.

Response:We absolutely agree that this is only the first study, the results should be considered as preliminary ones, and a multi-year trend or at least a repeat of the winter trend is needed. We did not expect that a minor soil-related yeast species, Tausonia pullulans, would show such a response to developing in an ornithogenic soil at sub-freezing temperatures. And also that many strains would prove to be active producers of cold-adapted enzymes. All the strains have been preserved in the collection for further study. But if the pattern discovered is repeated, it appears that a source has been found to isolate a large numbers of strains of biotechnologically important species at minimal cost. In our humble opinion, there is a perspective here.

Reviewer 3 Report

I think it can be accepted now.

I think it can be accepted now.

Author Response

Many thanks to the Reviewer for the work, the important comments and the time. Thanks to your work we were able to improve the manuscript.